# Attention deficit in children with attention deficit hyperactivity disorder at primary school age measured with the attention network test (ANT): A protocol for a systematic review and meta-analysis

**Gabriela Gradys**[1,2]ᴼ*, **Małgorzata Lipowska**[1,2], **Łucja Bieleninik**[1,3]ᴼ, **Angel M. Dzhambov**[4,5]ᴼ

1 Institute of Psychology, Faculty of Social Sciences, University of Gdansk, Gdańsk, Poland, 2 Institute of Applied Psychology, Faculty of Management and Social Communication, Jagiellonian University, Kraków, Poland, 3 GAMUT-The Grieg Academy Music Therapy Research Centre, NORCE Norwegian Research Centre, Bergen, Norway, 4 Department of Hygiene, Faculty of Public Health, Medical University of Plovdiv, Plovdiv, Bulgaria, 5 Institute for Highway Engineering and Transport Planning, Graz University of Technology, Graz, Austria

ᴼ These authors contributed equally to this work.
* gabriela.gradys@gmail.com

## Abstract

### Background

Attentional deficits are among the most bothersome symptoms of attention deficit hyperactivity disorder (ADHD). To date, the neurological basis of attentional deficits has not been fully described according to the diagnostic criteria. ADHD may result from deficits in various attributes of attention. There is no specialist neuropsychological diagnostic method that allows reliable distinction between primary attention disorders in the etiology of ADHD and secondary problems that may arise due to co-morbidities. This protocol aims to systematically review the literature to evaluate patterns of attention common to school-age children either diagnosed with ADHD or at high risk of ADHD, as measured by the neuropsychological attention network test (ANT).

### Methods

Our search strategy will consist of electronic databases (PubMed, PsychInfo, Web of Science, EMBASE, and Cochrane Library) and hand searching. Both prospective cohort studies and prospective studies of intervention effects will be included, provided they used the ANT. The primary output variable will be attention deficits. Screening and eligibility will be done independently by two reviewers based on pre-specified eligibility criteria. Data extraction will be based on a pre-pilot data extraction form and conducted by two authors independently. The risk of bias will be assessed by two authors independently. The rating of the certainty of the entire body of evidence will be evaluated using the GRADE approach. Any discrepancies identified at any stage of the review will be resolved by discussion or/and

**Data Availability Statement:** The article does not report data and the data availability policy is not applicable.

**Funding:** The "NeuroSmog: Determining the impact of air pollution on the developing brain" project is carried out within the TEAM-NET program of the Foundation for Polish Science co-financed by the European Union under the European Regional Development Fund (Nr. POIR.04.04.00-1763); Statutory funds of the Institute of Psychology of the University of Gdańsk. The funding body does not influence the design of the study and the writing of the manuscript.

**Competing interests:** The authors have declared that no competing interests exist.

**Abbreviations:** ADD, Attention Deficit Disorder; ADHD, Attention-deficit/hyperactivity disorder; ANT, Attention Network Test; APA, American Psychological Association; CD, Conduct Disorder; ICD, International Classification of Diseases; MeSH, Medical Subject Headings; NOS, The Newcastle-Ottawa Scale; ODD, Oppositional Defiant Disorder; PRISMA, Preferred Reporting Items for Systematic Reviews and Meta-Analysis; Rob 2, Revised Cochrane Collaboration Risk of Bias Tool; ROBINS-I, Risk of Bias in Non-randomized Studies - of Interventions.

consultation with another reviewer. We plan a narrative synthesis of findings and a quantitative meta-analysis if the data allow.

## Discussion

The research will identify patterns of neuropsychological ANT results characteristic of both school-age children diagnosed with ADHD and those at high risk of having ADHD. Our results could be used to check whether the pattern of a child's performance in the ANT corresponds to the characteristic pattern of the results of children with ADHD. At present, the ANT is used only in research; the results of this review will serve as a useful benchmark. Hopefully, in the future, it will be possible to use the ANT in the wider diagnosis of ADHD.

## Systematic review registration

PROSPERO: CRD42021249768.

## Introduction

Attention-deficit hyperactivity disorder (ADHD) is a neurodevelopmental disorder, including attention deficit disorder (ADD) and/or hyperactivity and impulsiveness [1]. It is one of the most common neurodevelopmental disorders occurring in school-aged children, typically being noticed between six and nine years of age, and symptoms may continue into adulthood [2]. The prevalence of ADHD is estimated to be between 6.7 and 7.8% [3]. Consequently, in a typical class of 25–30 children, between 1 and 3 pupils may have ADHD symptoms [4]. Pupils with ADHD typically underachieve academically relative to non-ADHD classmates [5], they are more likely to repeat a grade, be referred for special education services, be suspended from school, and drop out of school relative to students without disabilities [6]. These studies, as mentioned above, indicate that well-functioning attention seems to be a prerequisite for achieving good school results. Furthermore, ADHD has a crucial influence on other areas of life, both in childhood and adulthood. Children with ADHD more often than their normally developing peers experience difficulties in relations with their parents and siblings [7]. Parents of children suffering from ADHD more often experience depression and dissatisfaction with their role than parents of children without this disorder [7]. In relationships, the ADHD children experience more often disappointment by being excluded from the group or by making a smaller number of friends compared to normally developing peers. Additionally, friendships they make tend to last shorter [8]. Difficulties experienced in so many areas lead to lowered self-esteem, problems with behavior and coping with emotions and to the difficulties in adult life such as being dismissed from employment, having interpersonal issues with their colleagues, increased risk of drugs and substance abusing or increased chance of having other mental diseases [7].

For those diagnosed with ADHD, attention deficits are among the symptoms that have the most negative impacts on their everyday lives. The neurological basis for the occurrence of ADHD has not yet been fully described [9]. The diagnostic criteria proposed by the American Psychological Association [1] are written in such a way that suggests that ADHD may result from deficits of various attention attributes. No neuropsychological methods exist in specialist diagnostics that allow reliable distinctions to be made between primary attention disorders in ADHD and secondary problems that may arise due to other disorders, such as specific learning

disorders or behavioral disorders like conduct disorder (CD) and oppositional defiant disorder (ODD) [10–12]. Primary attention deficits are caused by abnormal brain development, like in ADHD. Secondary attention deficits can appear during the lifespan as a result of unhealthy habits, such as suboptimal nutrition [13], long screen time or poor sleeping; other neurodevelopmental and psychiatric disorders [14, 15], challenging life experiences [16, 17], and brain injury [18]. The distinction between primary and secondary attention deficits is fundamental with respect to selecting relevant therapeutic approaches, pharmacological treatments, or combinations of these. At present, few clinical measurement tools exist to diagnose attention deficits in children (e.g., Continuous Performance Test, Child Behavior Checklist), and there are no methods available for the diagnosis of primary attention deficit. The neuropsychological attention network test (ANT) may be useful in this capacity, but efforts are needed to investigate the performance of the test more carefully [19].

## Description of the method

The ANT is a neuropsychological test based on attention network theory [20] which describes an attention system as comprising three networks:

1. Orienting network—relating to the ability to maintain increased sensitivity to new, upcoming stimuli.

2. Alerting network—concerning the ability to select stimuli and focus attention on the person's stimulus of interest.

3. Conflict or executive network—reflecting the ability to control a behavioral response in response to a stimulus that enables two alternative responses.

The ANT is a computer-based graphical tool that can be used to measure the efficiency of each of these networks by collecting information on the degree of correctness and the reaction time of participant responses to presented stimuli [21]. In the test, participants are given various visual cues and/or warning tones while they attempt to indicate the direction in which a target arrow is pointing by pressing arrow keys on a keyboard. Participants are shown one of three types of flankers (neutral, compliant, and inconsistent) preceded by one of four types of clues (none, double, center, and spatial). During the whole test, participants are shown a small cross at the center of the screen. There are instructed to look at them all the time. This is a fixation point, used mostly between the clue and the flanker, which makes their eyes relatively stationary. In this way, they achieve retinal stabilization, thanks to which the presented figure (clue or flanker) tends to rapidly fade away. Studies can be conducted using different rules, e.g., in the presentation time of the fixation, signal, target, and final fixation slides. The target presented can be fish (in the child version) [22] rather than an arrow (in the adult version) [21]. Fig 1 illustrates the test.

The numbers under each slide are a common presentation range. In the original ANT procedure [21], each slide was shown for a random amount of time between 400 and 1600 msec for the first fixation, 100 msec for a cue, 400 msec for fixation of a cue, <1700 msec for a target and 3500 msec for the last fixation minus first fixation period and reaction time for a target. In the child version of ANT, times have been changed to adapt the original ANT [21], to children's abilities—slide timings were similar for first fixation slides and target, 150 msec for a cue, 450 msec for cue's fixation, 2000 msec of feedback and 1000 msec for finish fixation. Importantly, several versions of the ANT have been developed over the past two decades (for example ANT-C [22], LANT [23], ANT-R [24], ANT-I [25], ANTI-V [26]). These differ from the original version in the terminology used, and in certain aspects and characteristics of the tasks, but in a general measure the same outcome.

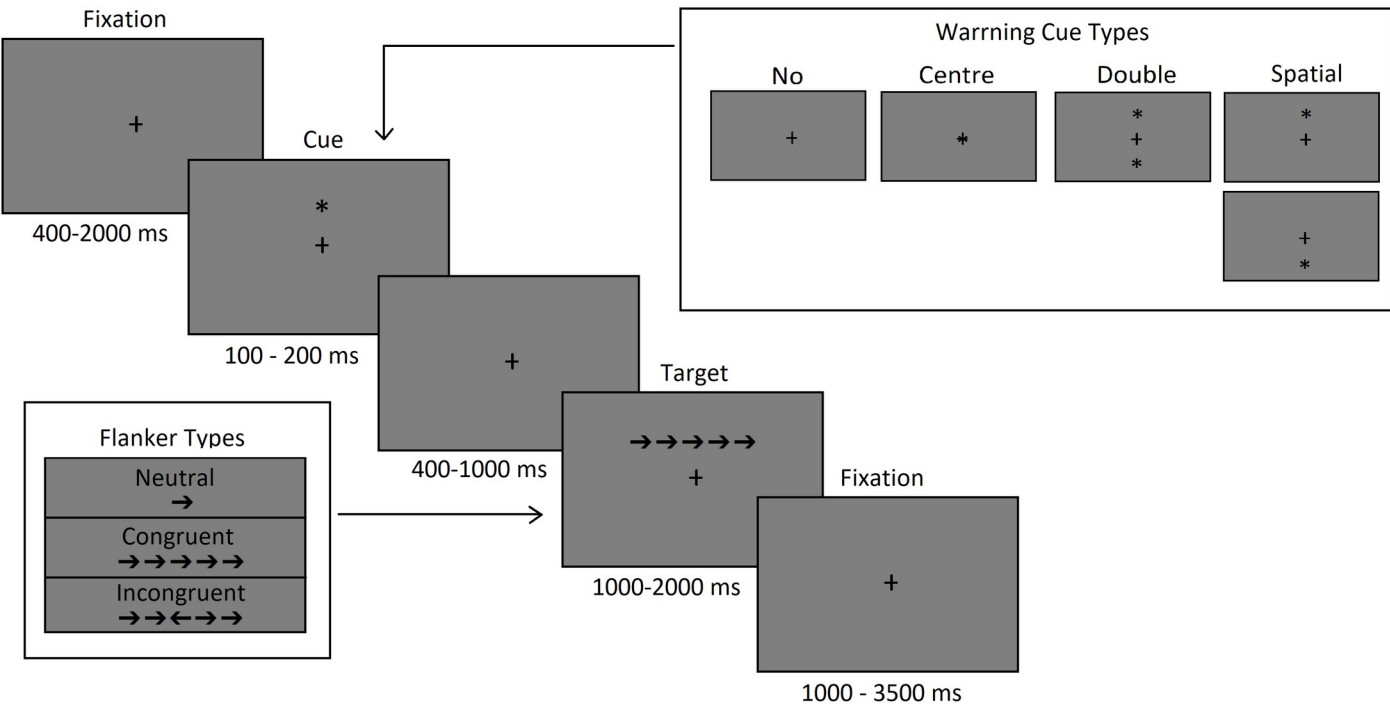

**Fig 1. Attention network task protocol.** Based on the original ANT version [21] and another previous research [19, 22].

## Previous research and the rationale for the review

To the best of our knowledge, little research has been done on the measurement of attention deficit in children with ADHD using the ANT. Attention deficits in children with ADHD do not necessarily affect the entire spectrum of attention. They may only relate to specific features of attention. Through careful examination of deficits in the three attention networks, it is possible to identify exactly which areas of attention are impaired in children with ADHD and thus allow for the implementation of more effective treatment. To date, a single narrative review paper [19], and a single meta-analysis paper [27] have been undertaken on this topic. However, neither paper followed the PRISMA Statement [28]. The narrative review by Vázquez-Marrufo, García-Valdecasas Colell [19] is based on 15 studies of changes in attention network measured with the ANT in school-age children and adults with ADD. The authors reported that the results of those studies were inconsistent. Some of the study results reported lower efficiency in all three attention networks in ADD children, but most of them found evidence for differences only in the alerting and control networks. Some of including research did not find any differences between the control and ADD groups. That may be caused by different ANT procedures or analytical methods. In a very recent publication on the ANT, Arora, Lawrence, and Klein [27] developed an ANT database containing all studies that used the ANT as of 2019 and conducted two meta-analyses to explore task performance in children with and without ADHD. The authors of that paper found a significant difference between executive and alerting networks between children with and without ADHD. However, there were limitations in their methodology. Firstly, children with ADHD diagnosis and those exhibiting a high risk of ADHD were not analyzed separately but combined into one group. In addition, their meta-analysis did not encompass all possible results of the ANT as analyses only incorporated overall reaction time and attention networks. The number of mistakes made by participants was not considered; thus, some questions still need to be answered. Importantly, the authors included only those

studies that followed the original ANT method [21], which strongly impacted the results obtained. Another major source of uncertainty is that the bias of included studies and the overall certainty of the evidence was not formally assessed, hindering the degree to which overall findings could be critically evaluated. Overall, there is abundant room for further analyses regarding the ANT in children with ADHD. Notably, it is unclear whether a child with ADHD performs the ANT differently from a child without ADHD, and if so, in which outcomes these differences would manifest themselves, and in which direction any changes would be observed.

## Aim

In this review, we aim to establish a single, distinctive pattern of ANT performance results for ADHD children, including the efficiency of attention network, reaction time, and numbers of all types of errors, none of which have been evaluated so far. This will allow comparison of ANT results obtained in future experimental studies are consistent with the pattern observed in ADHD respondents, or whether they are the result of the heterogeneity of the study group. This knowledge will allow the use of the ANT in further research and perhaps serve as a basis for the test's later standardization, which could make the ANT useable for broad diagnoses of attention. The last motivation to conduct the current systematic review was to provide an more current analysis of reaction time and attention networks, because the only meta-analysis which has been published included papers as of 2019 [27]. In summary, we identified a need for a rigorous systematic review with meta-analysis of the ANT to evaluate patterns of attention deficit characteristic of school-age children diagnosed with ADHD and those at high risk of ADHD.

This systematic review seeks to address the research question: Do primary school-age children diagnosed with ADHD or at risk of ADHD demonstrate differences in the efficiency of functioning of the three attention networks measured with the ANT when compared to non-ADHD children? In answering this question, we will refer to differences in performance of children diagnosed with ADHD and at risk of ADHD in terms of alarm, conflict, and orientation networks compared to children without any ADHD symptoms. The following secondary questions were posited:

1. Do primary school-age children diagnosed with ADHD or at risk of ADHD, compared to non-ADHD children, demonstrate differences in correctness scores (number of commission and omission) measured with the ANT?

2. Do primary school-age children diagnosed with ADHD or at risk of ADHD, compared to non-ADHD children, demonstrate differences in the number of mistakes made in a particular type of clue or flankers measured with the ANT?

3. Do primary school-age children diagnosed with ADHD or at risk of ADHD, compared to non-ADHD children, demonstrate differences in the ratio times in a particular type of clue or flankers measured with the ANT?

4. Do primary school-age children diagnosed with ADHD or at risk of ADHD, compared to non-ADHD children, demonstrate differences in results of the ANT depending on the used version of ANT, e.g. ANT-C [22], LANT [23], ANT-R [24], ANT-I [25], ANTI-V [26]?

## Methods

We will undertake a comprehensive literature search following guidelines outlined in the Preferred reporting items for systematic review and meta-analysis protocols (PRISMA-P) [29]. Our protocol was registered with the International Prospective Register of Systematic Reviews (PROSPERO) database (registration number: CRD42021249768). All the other parts of the

systematic review and the final article will be prepared based on the guidelines outlined in the Preferred reporting items for systematic review and meta-analysis (PRISMA) [30].

## Eligibility criteria

Studies will be selected according to the criteria outlined below.

**Participants.**   We will include studies of individuals of primary school age, which according to the assumptions of The International Standard Classification of Education 2011 means the age between 5 and 13 years [31], both sexes without restriction to nationality, who had an ADHD diagnosis or were considered to be at high risk of ADHD. ADHD should have been diagnosed by a specialist (psychiatrist, clinical psychologist, or any other qualification medical staff) based on DSM-5 (ADHD diagnostic code: 314) [1] either earlier versions of the DSM or the International Classification of Diseases (ICD, ADHD diagnostic code according to the ICD-11: 6A05) [32]. The high-risk group of ADHD includes children with symptoms of ADHD measured by ADHD symptoms questionnaires (like CONNERS-3, Structured Diagnostic Interview Questionnaire for ADHD—or foreign equivalents). Children with comorbidities (such as anxiety disorder, conduct disorder, learning disorder, and oppositional defiant disorder) will be included due to the nature of common coexistence with ADHD (8–10). However, other atypical concomitant or concurrent disorders (such as eating disorders, depressive or bipolar disorders, obsessive-compulsive disorders, or factitious disorders) will be excluded because of the potential contribution to the clinical symptoms and influence of AND performance.

**Intervention/exposure.**   The review will include studies regardless of the type of intervention for which the effectiveness was measured using the ANT.

**Comparator/control.**   We will include studies including individuals of primary school age, both sexes, without restriction to nationality, without ADHD symptoms or ADHD diagnosis.

**Outcomes.**   Performance of any version of the ANT, e.g. ANT-C [22], LANT [23], ANT-R [24], ANT-I [25], ANTI-V [26].

*Primary outcomes.* Mean and standard deviation or median and range (or a standardized effect measures such as Cohen's d) of the executive, alerting, and orienting attention network, measured by the ANT.

*Secondary outcomes.* Mean and standard deviation or median and range and intra-individual variability (or a standardized effect measures such as Cohen's d) of general reaction time achieved in the ANT. A number of omissions (missing answers) and a number of commissions (wrong answers) errors or if there will be a lack of that data, general correctness rate (percent of the correct answer) reached in ANT.

**Study type.**   We will include prospective cohort studies addressing attention deficit measured with the ANT and prospective studies of intervention effects with a control group (both randomized and non-randomized controlled).

**Location.**   We do not impose any restrictions on the area of the conducted research. The target population could be recruited from both primary schools or health care facilities (such as primary care settings, therapeutic settings, or diagnostic settings).

## Search strategy

**Information sources.**   The following electronic databases will be searched: PubMed, PsychInfo, Web of Science, EMBASE, DARE, and the Cochrane Library. Electronic database searching will be supplemented by hand-searching reference lists of the included review articles to identify any additional studies. The search will be not restricted to any language, sample

size, or year of publication. We will exclude editorials, letters, case studies, case series, and conference abstracts.

**Search.**   Literature search strategies will be developed using Medical Subject Headings [33] or equivalent and text word terms and words related to the nosological unit and the ANT. In addition, Boolean operators along with proximity operators (parentheses and quotations) for each database will be applied. The search strategy included terms relating to condition (attention deficit disorder with hyperactivity [33] OR ADHD) and measurement tool (Attentional Network Test" OR "Attentional Networks Test" OR "Attentional Network Task" OR "Attentional Networks task" OR "Attention Network Test" OR "Attention Networks Test" OR "Attention Network Task" OR "Attention Networks task"). We have already piloted the initial search strategy (including searching terms and filters) for PubMed in March 2021 to investigate whether the searching strategy allows us to find potentially relevant reports [S1 Checklist]. The pre-tested searching allowed us also to improve the search terms.

### Data collection and analysis

**Study selection.**   One reviewer will search databases and hand search the reference list of the included review articles. All potentially relevant records will be extracted to EndNote reference management software [34]. At this stage, duplicates will be detected and deleted. Two review authors will screen titles and abstracts for their eligibility for inclusion under the above-defined criteria providing the reason(s) for rejection. Any discrepancies at the screening stage will be resolved by discussion with another reviewer. The eligibility criteria will be pre-tested on a reasonable sample of reports. We will obtain full reports for all titles that appear to meet the inclusion criteria or where there is any uncertainty. Two review authors independently will then screen the full-text reports and decide whether these meet the inclusion criteria proving the reason for rejection. Assessment of the relevance of studies will be conducted by clinical psychology researchers experts in the content area. They will seek additional information from each study's corresponding authors whenever it is necessary to resolve questions on eligibility. Eligibility criteria for each study will be assessed in order of importance, starting from participants, followed by the outcome, intervention/exposure, comparator/control, and study design. Applying this strategy causes the first 'no' response to be the primary reason for excluding the study, and the remaining criteria will not be assessed. We will record the reasons for excluding studies. We will use a formal measure of an agreement to describe the extent to which assessments by two authors are the same. Disagreements at the assessment eligibility stage will be resolved by discussion with another reviewer. Multiple records of the same study will be merged based on matching the following study characteristics: author names, location and setting, numbers of participants and baseline data, and duration of the study. Where any uncertainties remain, we will contact the corresponding authors. Neither of the review authors will be blind to the journal titles, the study authors, or their institutions. The PRISMA template [30] will produce a flow chart showing details of studies included and excluded at each stage of the study selection process.

### Data extraction and management

Two authors independently will extract data from the studies based on a specifically designed and pre-piloted data extraction form. Data extraction will be done by content area experts from the clinical psychology field who are familiar with the ANT, and who will be trained in how to code entries in the data collection form. Discrepancies will be resolved by discussion or/and consultation with another reviewer when needed. In case of discrepancies that cannot be resolved, we will contact the study authors; however, if this is unsuccessful, the

discrepancies will be reported in the review. Corresponding authors will also be contacted to obtain any missing data. In case of multiple records of the same study/project, we will extract data from each report separately and combine information across multiple data collection forms afterward.

The following information will be extracted from the studies:

- Publication details–title, author; year of publication; DOI number; country of a study conducting

- The number of participants per group

- Characteristics of the clinical population–age, sex, ADHD group type (ADHD/risk of ADHD), ADHD intensity evaluated by the results of the questionnaire (e.g., Conners 3); the sub-type of ADHD diagnosis (predominantly inattentive, predominantly hyperactive/impulsive, and combined), diagnosis provider; diagnosis method(s); comorbidities, pharmacotherapy (yes/no); pharmacotherapy used during ANT assessment (yes/no)

- Characteristics of the control population–age, sex

- Study design—Prospective cohort study/intervention study

- The ANT results—Mean and standard deviation or median and range (or a standardized effect measures such as Cohen's d) of the executive, alerting and orienting attention network, mean and standard deviation or median and range, as well as intra-individual variability of general reaction time. A number of omissions (missing answers) and a number of commissions (wrong answers) errors or if there will be a lack of that data, general correctness rate (percent of the correct answer). In observational studies with repeated measurement or intervention studies with several time points, we will always extract baseline data. The version of the ANT used, how the training of the ANT was performed, how the instructions were presented, the person conducting the test and their interventions with the child during the test, and any other descriptive data about the ANT performance and conducting.

- Characteristics of the interventions–types of intervention, frequency, duration.

## Assessment of the risk of bias in included studies

An assessment of potential bias will be performed independently by two review authors. Any discrepancies will be resolved by arbitration among reviewers, together with content area experts from clinical trial methodology if needed. To assess the risk of bias in each included study, we will use the Revised Cochrane Collaboration Risk of Bias Tool (RoB 2) for randomized trials [35] and the Risk Of Bias In Non-randomized Studies—of Interventions (ROBINS-I) tool for non-randomized studies [36]. The risk of bias will be judged as high, low, or unclear risk bias. For cohort studies, we will use The Newcastle-Ottawa Scale [37], which assesses the quality of cohort studies by a judgment of the selection of the study groups, the comparability of these groups, and the ascertainment of either the exposure or outcome of interest [38].

## Data analysis and synthesis

During the analysis a three factors will be considered:

- Correctness—measured by the number of omissions (missing answers) and a number of commissions (wrong answers) errors or if there will be a lack of that data, general correctness rate (percent of the correct answer) achieved by children.

- Reaction time—mean and standard deviation or median and range and intra-individual variability of general experiment reaction time.

- Attention network—measured by the difference in mean or median reaction times between:

  ○ Double clue vs no clue or tone vs no tone—for alerting network

  ○ Valid cue vs invalid clue—for orienting network

  ○ Congruent vs incongruent trial type—for executive network.

The three factors mentioned above will be compared between the ADHD clinical group and the control group as well as between the high-risk ADHD group and the control group to detect potential specific differences in children with ADHD to establish a characteristic pattern. We will conduct multiple meta-analyses, one for each version of the ANT.

During the analysis the fallowing characteristics will be compare:

- ADHD diagnosis vs high-risk of ADHD vs non-ADHD child–we will measure differences in the same factors of the ANT as described above (attention networks, correctness, and reaction time). It will be helpful to describe a characteristic pattern of attention disorder in ADHD and get an answer for the first, second, and third questions.

- We will compare all results obtained with all versions of the ANT with one another to answer the fourth secondary question. We will check whether the version of the ANT used affects the results obtained by all children. Furthermore, we will check whether they achieve similar effects, due to the fact that not every version of the ANT was designed to be used on children, but they are commonly used in that way. We will then be able to exclude these versions of the ANT, which introduced bias.

- We will compare type A ADHD vs type B ADHD to check whether the results of the ANT are different for each ADHD type. If such a difference were identified, it would be possible to predict the ADHD type based on the ANT results and to describe the pattern of the ANT results for every type of ADHD.

- We will compare symptoms of ADHD at intensity A against symptoms at intensity B to check whether the severity of the symptoms of attention disorders is reflected in the results of the ANT. Were this found to be the case, it would be possible to predict attention disorders and their severity based on the results of the ANT. Children with ADHD diagnoses are a heterogeneous group with different intensities of attention deficit. Even in the DSM 5 diagnostic criteria, a child should obtain a minimum of 6 out of 9 criteria of inattention to gain an ADHD diagnosis. That analysis could show that children with ADHD could also have slightly different results in the ANT, but still, it would be possible to differentiate them from children without attention deficits.

We will make a narrative synthesis of the findings from the included studies, structured around the participants (demographic and clinical characteristics), the ANT results, and characteristics of the interventions (in the case of intervention-based research), along with a comparative table. Quantitative data will be combined only if means and standard deviations are available or can be derived from available data.

If two or more studies are found to be sufficiently clinically and statistically homogeneous to be combined in a meta-analysis, we will pool the effect estimates of these studies using standard meta-analytical techniques described in the Cochrane Handbook for Systematic Reviews of Interventions [39]. We will employ the fixed-effects estimator in the absence of materially important heterogeneity and the DerSimonian-Laird random effects estimator otherwise.

We will only combine studies for which the effect estimates can be converted to a common metric (e.g., group means and standard deviations into Hedges's g). We will not combine in the same meta-analysis studies of different design (observational and intervention studies). We will also not pool together multiple estimates coming from the same study or from statistical tests based on the same or overlapping subjects, as they cannot be considered independent, without taking within-study correlation into consideration [40, 41]. If a study reports effect estimates for the same outcome from more than one between-group tests, we will extract the estimate we believe provides more direct evidence or is less biased (e.g., is associated with a larger sample size or better adjustments) and will justify our decision to prioritize it.

Statistical heterogeneity in the models will be suggested by a significant Cochran's Q at the $p < 0.1$ level and quantified by the $I^2$ statistic as follows: mild ($< 30\%$), moderate ($30–50\%$) or high ($> 50\%$) [42]. We will also inspect the direction of individual study effect sizes and the overlap of their confidence intervals.

Presence of publication bias for each outcome will be judged graphically and quantitatively. If the meta-analysis includes 10 or more studies, we will construct a funnel plot and test for asymmetry using Egger's regression test [43]. Next, we will also generate a Doi plot, which plots study-level effect sizes against a rank-based measure of precision (z-score, where the midpoint is defined by the most precise studies and the less precise studies are scattered outward towards the tails of the plot) [44]. Asymmetry in a Doi plot can be tested with even 5–10 studies using an index called Luis Furuya-Kanamori (LFK) index, which is not a p-value based test, rather quantifies the difference between the two areas under the Doi plot curve created by the midpoint [45]. Major asymmetry will be indicated by an asymmetrical Doi plot and LFK index $>|2|$ [44].

Quantitative synthesis will be carried out using Stata v. 17 (College Station, TX: StataCorp LP.) and MetaXL v. 5.3 (EpiGear International Pty Ltd, Sunrise Beach, Queensland, Australia).

## Sensitivity analyses

Where applicable, we will conduct leave-one-out meta-analysis for each outcome to determine whether excluding studies one-at-a-time would materially change the pooled effect. This way we could identify influential studies.

Given growing concerns about the appropriateness of the random effects model and its potential to yield overly liberal findings, we will also re-run the meta-analysis using the inverse-variance heterogeneity model; it was built under the fixed effect model assumption with a quasi-likelihood based variance structure to retain a correct coverage probability and yield more conservative pooled estimates regardless of heterogeneity [46, 47].

## Overall quality of evidence

For each outcome, we will grade the quality of evidence using the Grading of Recommendations, Assessment, Development and Evaluations (GRADE) approach [41, 48]. Evidence will be judged as "high", "moderate", "low", or "very low" quality depending on the extent to which we can be certain that the pooled effect estimate is close to the true effect. For randomized trials, we will start at "high", and for observational studies, at "moderate" quality. The quality of evidence will be downgraded by 1 level for each of the following reasons–high risk of bias across the studies, indirectness of evidence (indirect population, intervention, control, outcomes), high heterogeneity ($I^2 > 50\%$) or inconsistency of results across studies, imprecision of results (wide confidence intervals, small sample size), and high probability of publication bias (presence of meaningful funnel plot and/or Doi plot asymmetry). Since there are no clear-cut recommendations on imprecision with continuous outcomes and standardized effect measures, we will downgrade if the sample size is $< 620$ (calculated under standard

assumptions of $\alpha = 0.05$, power = 0.80, and effect size of 0.20) [48], and will consider the width of the confidence interval around the point estimate and the range of values it includes.

If there are no serious concerns about risk of bias, we may upgrade the quality of evidence by one level for large magnitude of effect, if a dose-response gradient is observed, and/or if accounting for all plausible confounding would reduce the pooled effect or suggest a spurious effect when results show no effect [48]. We define a large effect according to Cohen's convention of 0.80 [49], although we recognize that this cutoff may be too strict or not equally relevant across psychological sub-disciplines [50, 51]. In addition, to upgrade for a large effect, the lower limit of the confidence interval of the point estimate will have to be at least 0.80 or greater [48].

## Discussion

We anticipate that our review will help future researchers using the ANT to find a reliable benchmark for observed results and that it will contribute to furthering research into the possibility of using the ANT for a more comprehensive diagnosis of attention disorders in children, including ADHD. We will update and provide a critical apprisal of the body of evidence on ANT. It may be possible that the characteristic patterns of the ANT performed by children with ADHD, compared with their counterparts, will be used as a reference point in comparative subjects to determine whether a given intervention (psychological, pharmacological, or other) used to reduce attention deficits in children is effective and results in actual, statistically significant differences in the efficiency of the attention networks of these children.

## Supporting information

**S1 Checklist. The completed PRISMA-P checklist.**
(DOC)

## Acknowledgments

We would like to thank James Grellier for reviewing our manuscript for linguistic correctness.

## Author Contributions

**Conceptualization:** Gabriela Gradys, Łucja Bieleninik.

**Data curation:** Gabriela Gradys, Angel M. Dzhambov.

**Formal analysis:** Angel M. Dzhambov.

**Funding acquisition:** Małgorzata Lipowska.

**Investigation:** Gabriela Gradys.

**Methodology:** Łucja Bieleninik, Angel M. Dzhambov.

**Project administration:** Gabriela Gradys.

**Resources:** Małgorzata Lipowska, Łucja Bieleninik.

**Software:** Małgorzata Lipowska.

**Supervision:** Małgorzata Lipowska, Łucja Bieleninik, Angel M. Dzhambov.

**Validation:** Małgorzata Lipowska, Łucja Bieleninik, Angel M. Dzhambov.

**Visualization:** Gabriela Gradys.

**Writing – original draft:** Gabriela Gradys.

**Writing – review & editing:** Małgorzata Lipowska, Łucja Bieleninik, Angel M. Dzhambov.

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
