## [Decision Letter · Decision Letter 0]

20 Jun 2022

PONE-D-22-06141Attention deficit in children with attention deficit hyperactivity disorder at primary school age measured with the attention network test (ANT): a protocol for a systematic review and meta-analysisPLOS ONE

Dear Dr. Gradys,

Thank you for submitting your manuscript to PLOS ONE. After careful consideration, we feel that it has merit but does not fully meet PLOS ONE’s publication criteria as it currently stands. Therefore, we invite you to submit a revised version of the manuscript that addresses the points raised during the review process.

We look forward to receiving your revised manuscript.

Kind regards,

Gabriel G. De La Torre

Academic Editor

PLOS ONE

Journal Requirements:

Reviewers' comments:

Reviewer's Responses to Questions

**Comments to the Author**

1. Does the manuscript provide a valid rationale for the proposed study, with clearly identified and justified research questions?

Reviewer #1: Yes

Reviewer #2: Yes

2. Is the protocol technically sound and planned in a manner that will lead to a meaningful outcome and allow testing the stated hypotheses?

Reviewer #1: Yes

Reviewer #2: Partly

3. Is the methodology feasible and described in sufficient detail to allow the work to be replicable?

Reviewer #1: Yes

Reviewer #2: Yes

4. Have the authors described where all data underlying the findings will be made available when the study is complete?

Reviewer #1: Yes

Reviewer #2: Yes

5. Is the manuscript presented in an intelligible fashion and written in standard English?

Reviewer #1: Yes

Reviewer #2: Yes

6. Review Comments to the Author

You may also provide optional suggestions and comments to authors that they might find helpful in planning their study.

Reviewer #1: The present manuscript deals with a current and relevant theme in the world scenario; the work is well written and can be published in this journal. However, I suggest to the authors some notes that can improve the quality of the manuscript.

At the end of the introduction, I recommend inserting the research hypotheses more clearly because they are not in the text. Don't forget to bring them again to the "discussion" topic.

Regarding the method, although the authors have inserted the inclusion criteria for the participants, you also need to insert the exclusion criteria. It was unclear to me how the participants were recruited, which needs to be presented in the text.

The authors could also describe the data analysis in more detail, as some statistical components are important for the readers to understand these analyses' steps.

Reviewer #2: This is a well-structured protocol for a meta-analysis seeking to identify if there is an ANT performance pattern that distinguishes between ADHD children and neurotypical children, and to add options for ADHD diagnosis, which is relevant and necessary. The research questions are clearly justified in the introduction, and their proposed methods include Cochrane and PRISMA guidelines. There are some details in the planned analysis and outcomes description that should be addressed to improve this protocol.

Participants:

- What about control groups? Should studies include control groups? Studies without a control group, would be included? Are there any criteria for control groups?

Outcomes:

- Line 191: Is attention deficit an outcome? Or ANT performance.

-There seem to be three groups of measures: performance indexes, reaction times, and errors (or correctness). However, the same outcomes are named in different ways in the protocol (reaction time ratio/average number of reaction time, errors/mistakes) and it makes a bit difficult to follow general ideas. I recommend selecting a specific name for each measure and use the same name along the manuscript to improve readability.

Search Strategy:

-  Line 208:Will meeting abstracts be included?

Data extraction and management

 - The number of participants per group should also be extracted

- Line 257: Clinical characteristics of the population: will ADHD symptoms be extracted?

Data synthesis:

- Line 85: Authors could also contact the studies authors when means and standard deviations are not available.

Data analysis:

- This section is similar to the outcome description. There are just a few lines about planned analyses (301-302). This could be described in more detail, will effect sizes be computed?, random or fixed models will be used? how heterogeneity will be assessed? how meta-bias will be measured? Will funnel plots be used? In the case of significant publication bias, what will be implemented?

-What is the plan in case quantitative analyses are not possible?

7. PLOS authors have the option to publish the peer review history of their article (what does this mean?). If published, this will include your full peer review and any attached files.

Reviewer #1: No

Reviewer #2: **Yes: **Yaira Chamorro

---

## [Author Response · Author response to Decision Letter 0]

5 Aug 2022

PONE-D-22-06141

RESPONSE TO REVIEWER COMMENTS

TITLE: Attention deficit in children with attention deficit hyperactivity disorder at primary school age measured with the attention network test (ANT): a protocol for a systematic review and meta-analysis

GENERAL RESPONSE: Thank you for taking the time to thoughtfully complete this review. We appreciate the helpful and insightful comments from our two reviewers. We respond individually to the reviewers below.

Journal Requirements:

RESPONSE: Done as requested. 

RESPONSE: Thank you for this comment. The following statement was added in the revised manuscript (page 2, lines 44-45) as well as to the “Ethics Statement” field of the submission form: “The systematic review will use data that already exists and doesn’t require seeking institutional ethics approval or participant consent before commencing a systematic review”.

 

RESPONSE: Thank you for this comment. The following statement was added in the revised manuscript (page 2, lines 40-41) as well as to the “Data Availability statement” field of the submission form: “Deidentified research data will be made publicly available when the study is completed and published.”

RESPONSE: Thank you for this comment. Done as requested.

RESPONSE: Done as requested. 

 

REVIEWER 1

Reviewer #1: The present manuscript deals with a current and relevant theme in the world scenario; the work is well written and can be published in this journal. However, I suggest to the authors some notes that can improve the quality of the manuscript.

RESPONSE: Thank you for the evaluation of our manuscript and your useful comments. Detailed responses to your comments are listed below.

At the end of the introduction, I recommend inserting the research hypotheses more clearly because they are not in the text. Don't forget to bring them again to the "discussion" topic.

RESPONSE: We thank the reviewer for this comment. However, since we describe a systematic review protocol, we were required to follow the guidelines outlined in the Preferred reporting items for systematic review and meta-analysis protocols (PRISMA-P), where, rather than a hypothesis, review authors are required to provide the question(s) the review will address concerning participants, interventions, comparators, and outcomes. According to these guidelines, we believe it would be redundant to also provide a testable hypothesis (compared to the primary research). 

Regarding the method, although the authors have inserted the inclusion criteria for the participants, you also need to insert the exclusion criteria. It was unclear to me how the participants were recruited, which needs to be presented in the text.

RESPONSE: We thank the reviewer for this valuable comment. Following the PRISMA-P the statement of the eligibility criteria should specify the study characteristics to be used as criteria for eligibility for the review. We will include studies of individuals of primary school age, both sexes without restriction to nationality, who had an ADHD diagnosis or were considered to be at high risk of ADHD (page 9, lines 280-288); thus, anyone who does not meet these inclusion criteria will be excluded. To address this comment the following statement was added in the revised manuscript: “Children with comorbidities (such as anxiety disorder, conduct disorder, learning disorder, and oppositional defiant disorder) will be included due to the nature of common coexistence with ADHD (8-10). However, other atypical concomitant or concurrent disorders (such as eating disorders, depressive or bipolar disorders, obsessive-compulsive disorders, or factitious disorders) will be excluded because of the potential contribution to the clinical symptoms and influence of AND performance.” (Page 9, lines 288-293);

In terms of recruitment, we do not impose any restrictions on the area of the conducted research (page 10, lines 312-314). We agree that our description doesn’t look comprehensive. To address this comment the following statement was added: “The target population could be recruited from both primary school or health care facilities (such as primary care settings, therapeutic settings, or diagnostic settings).“

The authors could also describe the data analysis in more detail, as some statistical components are important for the readers to understand these analyses' steps.

RESPONSE: We thank the reviewer for this valuable comment. To address this gap, we decided to include Prof. Angel Dzhambov as a co-author of the paper to support us. This inclusion was discussed with and has received consent from all the co-authors of the paper. The following statements were added in the revised manuscript in terms of this comment: (page 15-16; lines 485-534)

“We will make a narrative synthesis of the findings from the included studies, structured around the participants (demographic and clinical characteristics), the ANT results, and characteristics of the interventions (in the case of intervention-based research), along with a comparative table. Quantitative data will be combined only if means and standard deviations are available or can be derived from available data. 

If two or more studies are found to be sufficiently clinically and statistically homogeneous to be combined in a meta-analysis, we will pool the effect estimates of these studies using standard meta-analytical techniques described in the Cochrane Handbook for Systematic Reviews of Interventions [39]. We will employ the fixed-effects estimator in the absence of materially important heterogeneity and the DerSimonian-Laird random effects estimator otherwise. 

We will only combine studies for which the effect estimates can be converted to a common metric (e.g., group means and standard deviations into Hedges’s g). We will not combine in the same meta-analysis studies of different design (observational and intervention studies). We will also not pool together multiple estimates coming from the same study or from statistical tests based on the same or overlapping subjects, as they cannot be considered independent, without taking within-study correlation into consideration [40, 41]. If a study reports effect estimates for the same outcome from more than one between-group tests, we will extract the estimate we believe provides more direct evidence or is less biased (e.g., is associated with a larger sample size or better adjustments) and will justify our decision to prioritize it.

Statistical heterogeneity in the models will be suggested by a significant Cochran’s Q at the p < 0.1 level and quantified by the I2 statistic as follows: mild (< 30%), moderate (30–50%) or high (> 50%) [42]. We will also inspect the direction of individual study effect sizes and the overlap of their confidence intervals. 

Presence of publication bias for each outcome will be judged graphically and quantitatively. If the meta-analysis includes 10 or more studies, we will construct a funnel plot and test for asymmetry using Egger’s regression test [43]. Next, we will also generate a Doi plot, which plots study-level effect sizes against a rank-based measure of precision (z-score, where the midpoint is defined by the most precise studies and the less precise studies are scattered outward towards the tails of the plot) [44]. Asymmetry in a Doi plot can be tested with even 5-10 studies using an index called Luis Furuya-Kanamori (LFK) index, which is not a p-value based test, rather quantifies the difference between the two areas under the Doi plot curve created by the midpoint [45]. Major asymmetry will be indicated by an asymmetrical Doi plot and LFK index >|2| [44]. 

Quantitative synthesis will be carried out using Stata v. 17 (College Station, TX: StataCorp LP.) and MetaXL v. 5.3 (EpiGear International Pty Ltd, Sunrise Beach, Queensland, Australia). 

Sensitivity analyses 

Where applicable, we will conduct leave-one-out meta-analysis for each outcome to determine whether excluding studies one-at-a-time would materially change the pooled effect. This way we could identify influential studies 

Given growing concerns about the appropriateness of the random effects model and its potential to yield overly liberal findings, we will also re-run the meta-analysis using the inverse-variance heterogeneity model; it was built under the fixed effect model assumption with a quasi-likelihood based variance structure to retain a correct coverage probability and yield more conservative pooled estimates regardless of heterogeneity [46, 47].”

 

REVIEWER 2

Reviewer #2: This is a well-structured protocol for a meta-analysis seeking to identify if there is an ANT performance pattern that distinguishes between ADHD children and neurotypical children, and to add options for ADHD diagnosis, which is relevant and necessary. The research questions are clearly justified in the introduction, and their proposed methods include Cochrane and PRISMA guidelines. There are some details in the planned analysis and outcomes description that should be addressed to improve this protocol.

RESPONSE: We thank you for your very constructive and insightful remarks on the manuscript. Detailed responses to your comments are listed below.

Participants:

- What about control groups? Should studies include control groups? Studies without a control group, would be included? Are there any criteria for control groups?

RESPONSE: We thank the reviewer for this comment. Following the PRISMA-P, the eligibility criteria we specified according to the PICOS model, thus both intervention and comparator/control group were described under the subsection entitled “Eligibility Criteria” following participants. Please see below:

“Comparator/Control: We will include studies including individuals of primary school age, both sexes, without restriction to nationality, without ADHD symptoms or ADHD diagnosis. (page 10, lines 299-300). Thus, in terms of this comment, no revision was provided. However, if the reviewer feels that this description is insufficient, we would be happy to provide further amendments. 

Outcomes:

- Line 191: Is attention deficit an outcome? Or ANT performance.

RESPONSE: We thank the reviewer for putting attention to this. We appreciate this valuable comment. The outcome is ANT performance as the reviewer suggested. To address this comment the statement was corrected in the revised manuscript (see page 10, lines 301-302) as fallow: “Outcomes: Performance of any version of the ANT, e.g. ANT-C [22], LANT [23], ANT-R [24], ANT-I [25], ANTI-V [26].”

 

-There seem to be three groups of measures: performance indexes, reaction times, and errors (or correctness). However, the same outcomes are named in different ways in the protocol (reaction time ratio/average number of reaction time, errors/mistakes) and it makes a bit difficult to follow general ideas. I recommend selecting a specific name for each measure and use the same name along the manuscript to improve readability.

RESPONSE: We thank the reviewer for this advice. The ANT is a complex measurement tool. To address this comment, we made all outcome names selfsame in the entire manuscript, as fallow: 

• Mean and standard deviation or median and range of the executive, alerting and orienting attention network, 

• mean and standard deviation or median and range and intra-individual variability of general reaction time. 

• A number of omissions (missing answers) and a number of commissions (wrong answers) errors or if there will be a lack of that data, general correctness rate (percent of the correct answer).

All changes were highlighted in the revised manuscript with track changes, please see pages 10, 13, 14, and 15.

Search Strategy:

- Line 208:Will meeting abstracts be included?

RESPONSE: We thank the reviewer for this comment. We aim to obtain full-length papers at the eligibility stage to gather the data required to plot in meta-analyses. However, screening is always based on a title and an abstract. In the case of our protocol, the search will be not restricted to any language as long as an English-written abstract will be available. Our searching strategy is provided mainly in electronic databases which all index the English version of the abstract; thus, to address this comment, we have decided to delete the following half of the sentence “[…] (provided an English language translation of the abstract is available) […]” in the revised manuscript. We hope this revision ensures that readers will not be confused by the relevant criteria of inclusion. 

 

Data extraction and management

 - The number of participants per group should also be extracted

- Line 257: Clinical characteristics of the population: will ADHD symptoms be extracted?

RESPONSE: We thank the reviewer for these comments. We apologize that this information was omitted so far. We fully agree that our description of data extraction and management should be supplemented. To reply to this, the following statements were added in the revised manuscript: “The number of participants per group” (page 12, line 388).” In terms of the second issue, yes, we aim to extract the information of the sub-type of ADHD diagnosis and ADHD intensity evaluated by the results of the questionnaire if available. 

In addition, when checking this information, we have realized that we have forgot to add also another points; therefore, this section was improved as follows (pages 12-13, lines 385-411): 

“The following information will be extracted from the studies: 

• Publication details – author; year of publication; DOI number; country of a study conducting 

• The number of participants per group

• Characteristics of the clinical population – age, sex, ADHD group type (ADHD/risk of ADHD), ADHD intensity evaluated by the results of the questionnaire (e.g., Conners 3); the sub-type of ADHD diagnosis (predominantly inattentive, predominantly hyperactive/ impulsive, and combined), diagnosis provider; diagnosis method(s); comorbidities, pharmacotherapy (yes/no); pharmacotherapy used during ANT assessment (yes/no)

• Characteristics of the control population – age, sex

• Study design - Prospective cohort study/intervention study

• The ANT results - Mean and standard deviation or median and range of the executive, alerting and orienting attention network, mean and standard deviation or median and range, as well as intra-individual variability of general reaction time. A number of omissions (missing answers) and a number of commissions (wrong answers) errors or if there will be a lack of that data, general correctness rate (percent of the correct answer). In observational studies with repeated measurement or intervention studies with several time points, we will always extract baseline data. The version of the ANT used, how the training of the ANT was performed, how the instructions were presented, the person conducting the test and their interventions with the child during the test, and any other descriptive data about the ANT performance and conducting.

• Characteristics of the interventions –types of intervention, frequency, duration.”

We hope that the revised paragraph will improve the quality of the data extraction reporting. However, if the reviewer feels that this description is insufficient, we would be happy to provide further amendments. 

Data synthesis:

- Line 85: Authors could also contact the studies authors when means and standard deviations are not available.

RESPONSE: We thank the reviewer for this comment. Yes, we agree with this advice. We had already added the statement “Corresponding authors will also be contacted to obtain any missing data” under the subsection entitled “Data extraction and management”. Thus, in terms of this comment, no revision was provided. 

Data analysis:

- This section is similar to the outcome description. There are just a few lines about planned analyses (301-302). This could be described in more detail, will effect sizes be computed?, random or fixed models will be used? how heterogeneity will be assessed? how meta-bias will be measured? Will funnel plots be used? In the case of significant publication bias, what will be implemented?

RESPONSE: We thank the reviewer for these valuable suggestions to improve this section. To address this gap, decided to include Prof. Angel Dzhambov as a co-author of the paper to support us, who added the following statements in the revised manuscript (page 15-16; lines 485-534):

“We will make a narrative synthesis of the findings from the included studies, structured around the participants (demographic and clinical characteristics), the ANT results, and characteristics of the interventions (in the case of intervention-based research), along with a comparative table. Quantitative data will be combined only if means and standard deviations are available or can be derived from available data. 

If two or more studies are found to be sufficiently clinically and statistically homogeneous to be combined in a meta-analysis, we will pool the effect estimates of these studies using standard meta-analytical techniques described in the Cochrane Handbook for Systematic Reviews of Interventions [39]. We will employ the fixed-effects estimator in the absence of materially important heterogeneity and the DerSimonian-Laird random effects estimator otherwise. 

We will only combine studies for which the effect estimates can be converted to a common metric (e.g., group means and standard deviations into Hedges’s g). We will not combine in the same meta-analysis studies of different design (observational and intervention studies). We will also not pool together multiple estimates coming from the same study or from statistical tests based on the same or overlapping subjects, as they cannot be considered independent, without taking within-study correlation into consideration [40, 41]. If a study reports effect estimates for the same outcome from more than one between-group tests, we will extract the estimate we believe provides more direct evidence or is less biased (e.g., is associated with a larger sample size or better adjustments) and will justify our decision to prioritize it.

Statistical heterogeneity in the models will be suggested by a significant Cochran’s Q at the p < 0.1 level and quantified by the I2 statistic as follows: mild (< 30%), moderate (30–50%) or high (> 50%) [42]. We will also inspect the direction of individual study effect sizes and the overlap of their confidence intervals. 

Presence of publication bias for each outcome will be judged graphically and quantitatively. If the meta-analysis includes 10 or more studies, we will construct a funnel plot and test for asymmetry using Egger’s regression test [43]. Next, we will also generate a Doi plot, which plots study-level effect sizes against a rank-based measure of precision (z-score, where the midpoint is defined by the most precise studies and the less precise studies are scattered outward towards the tails of the plot) [44]. Asymmetry in a Doi plot can be tested with even 5-10 studies using an index called Luis Furuya-Kanamori (LFK) index, which is not a p-value based test, rather quantifies the difference between the two areas under the Doi plot curve created by the midpoint [45]. Major asymmetry will be indicated by an asymmetrical Doi plot and LFK index >|2| [44]. 

Quantitative synthesis will be carried out using Stata v. 17 (College Station, TX: StataCorp LP.) and MetaXL v. 5.3 (EpiGear International Pty Ltd, Sunrise Beach, Queensland, Australia). 

Sensitivity analyses 

Where applicable, we will conduct leave-one-out meta-analysis for each outcome to determine whether excluding studies one-at-a-time would materially change the pooled effect. This way we could identify influential studies 

Given growing concerns about the appropriateness of the random effects model and its potential to yield overly liberal findings, we will also re-run the meta-analysis using the inverse-variance heterogeneity model; it was built under the fixed effect model assumption with a quasi-likelihood based variance structure to retain a correct coverage probability and yield more conservative pooled estimates regardless of heterogeneity [46, 47].”

 

-What is the plan in case quantitative analyses are not possible?

RESPONSE: We thank the reviewer for this valuable comment. If the quantitative analyses are not possible, we will only do data synthesis in a descriptive way. To clarify this, we revised the following sentence by changing “narrative” to “descriptive” as follows: “We will make a narrative synthesis of the findings from the included studies, structured around the participants (demographic and clinical characteristics), the ANT results, and characteristics of the interventions (in the case of intervention-based research), along with a comparative table. Quantitative data will be combined only if means and standard deviations are available or can be derived from available data” (page 15, lines 485-489).

Additional comments from authors:

(1) To address the comment about statistical methods from the reviewer 1st and 2nd, we have decided to include Prof. Angel Dzhambov as a co-author of the paper to support us. This inclusion was discussed with and has received consent from all the co-authors of the paper.

(2) To follow all items of PRISMA-P Statement we have also supplemented our revised manuscript with information about the confidence in cumulative evidence; therefore we described how the strength of the body of evidence will be assessed (page 17, lines 687-707).

Overall quality of evidence 

For each outcome, we will grade the quality of evidence using the Grading of Recommendations, Assessment, Development and Evaluations (GRADE) approach [41, 48]. Evidence will be judged as “high”, “moderate”, “low”, or “very low” quality depending on the extent to which we can be certain that the pooled effect estimate is close to the true effect. For randomized trials, we will start at “high”, and for observational studies, at “moderate” quality. The quality of evidence will be downgraded by 1 level for each of the following reasons – high risk of bias across the studies, indirectness of evidence (indirect population, intervention, control, outcomes), high heterogeneity (I2 > 50%) or inconsistency of results across studies, imprecision of results (wide confidence intervals, small sample size), and high probability of publication bias (presence of meaningful funnel plot and/or Doi plot asymmetry). Since there are no clear-cut recommendations on imprecision with continuous outcomes and standardized effect measures, we will downgrade if the sample size is < 620 (calculated under standard assumptions of α = 0.05, power = 0.80, and effect size of 0.20) [48], and will consider the width of the confidence interval around the point estimate and the range of values it includes. 

If there are no serious concerns about risk of bias, we may upgrade the quality of evidence by one level for large magnitude of effect, if a dose-response gradient is observed, and/or if accounting for all plausible confounding would reduce the pooled effect or suggest a spurious effect when results show no effect [48]. We define a large effect according to Cohen’s convention of 0.80 [49], although we recognize that this cutoff may be too strict or not equally relevant across psychological sub-disciplines [50, 51]. In addition, to upgrade for a large effect, the lower limit of the confidence interval of the point estimate will have to be at least 0.80 or greater [48]. 

(3) A couple of sentences were added in the background section in order to provide information about the implications of ADHD on later life. Please see lines 93-103 on page 4. We hope this information will supplement this section of the manuscript. 

(4) The following statement was added under the label entitled “Method” (lines 275-277, page 9): “All the other parts of the systematic review and the final article will be prepared based on the guidelines outlined in the Preferred reporting items for systematic review and meta-analysis (PRISMA) “

(5) The following statement was added under the Eligibility criteria, Participants section (lines 280-282, page 9) to clarify what we mean by primary school-age: “which according to the assumptions of The International Standard Classification of Education 2011 means the age between 5 and 13 years [31]”

(6) According to the advice of our new team member, a statistics specialist, we updated the primary and secondary outcome as follow (page 10, lines 104-109): “Primary outcomes: Mean and standard deviation or median and range (or a standardized effect measures such as Cohen’s d) of the executive, alerting, and orienting attention network, measured by the ANT. 

Secondary outcomes: Mean and standard deviation or median and range and intra-individual variability (or a standardized effect measures such as Cohen’s d) of general reaction time achieved in the ANT. A number of omissions (missing answers) and a number of commissions (wrong answers) errors or if there will be a lack of that data, general correctness rate (percent of the correct answer) reached in ANT.” and we add the sentence “Both types of studies will form separate groups in the meta-analysis.” Under the Eligibility criteria, study type (page 10, line 312).

(7) All typos were corrected.

(8) One reference (Rueda MR, Fan J, McCandliss BD, Halparin JD, Gruber DB, Lercari LP, et al. Development of attentional networks in childhood. Neuropsychologia. 2004;42(8):1029-40) was deleted due to the duplication.

(9) The following literature items were in terms of the aforementioned revisions:

a. Harpin VA. The effect of ADHD on the life of an individual, their family, and community from preschool to adult life. Archives of Disease in Childhood. 2005;90(suppl 1):i2-i7.

b. Lee Y, Mikami AY, Owens JS. Children’s ADHD Symptoms and Friendship Patterns across a School Year. Research on Child and Adolescent Psychopathology. 2021;49(5):643-56.

c. Borenstein M, Hedges LV, Higgins JPT, Rothstein HR. Introduction to Meta-Analysis. Introduction to Meta‐Analysis: John Wiley and Sons; 2009.

d. McKenzie JEB, S.E. Chapter 12: Synthesizing and presenting findings using other methods. In: Higgins J.P.T. TJ, Chandler J., Cumpston M., Li T., Page M.J., Welch V.A, editor. Cochrane Handbook for Systematic Reviews of Interventions. version 62 (updated February 2021) ed: Cochrane; 2021.

e. Borenstein M, Hedges LV, Higgins JPT, Rothstein HR. Introduction to Meta-Analysis. Introduction to Meta‐Analysis: John Wiley and Sons; 2009.

f. McKenzie JEB, S.E. Chapter 12: Synthesizing and presenting findings using other methods. In: Higgins J.P.T. TJ, Chandler J., Cumpston M., Li T., Page M.J., Welch V.A, editor. Cochrane Handbook for Systematic Reviews of Interventions. version 62 (updated February 2021) ed: Cochrane; 2021.

g. Higgins JP, Thompson SG. Quantifying heterogeneity in a meta-analysis. Stat Med. 2002;21(11):1539-58.

h. Egger M, Davey Smith G, Schneider M, Minder C. Bias in meta-analysis detected by a simple, graphical test. Bmj. 1997;315(7109):629-34.

i. Furuya-Kanamori L, Barendregt JJ, Doi SAR. A new improved graphical and quantitative method for detecting bias in meta-analysis. Int J Evid Based Healthc. 2018;16(4):195-203.

j. Furuya-Kanamori L, Xu C, Lin L, Doan T, Chu H, Thalib L, et al. P value-driven methods were underpowered to detect publication bias: analysis of Cochrane review meta-analyses. J Clin Epidemiol. 2020;118:86-92.

k. Doi SA, Barendregt JJ, Khan S, Thalib L, Williams GM. Advances in the meta-analysis of heterogeneous clinical trials I: The inverse variance heterogeneity model. Contemp Clin Trials. 2015;45(Pt A):130-8.

l. Doi SAR, Furuya-Kanamori L, Thalib L, Barendregt JJ. Meta-analysis in evidence-based healthcare: a paradigm shift away from random effects is overdue. Int J Evid Based Healthc. 2017;15(4):152-60.

m. Schünemann H BJ, Guyatt G, Oxman A,. RADE handbook for grading quality of evidence and strength of recommendations. . Updated October 2013. ed. guidelinedevelopment.org/handbook: The GRADE Working Group; 2013.

n. Cohen J. A power primer. Psychol Bull. 1992;112(1):155-9.

o. Schäfer T, Schwarz MA. The Meaningfulness of Effect Sizes in Psychological Research: Differences Between Sub-Disciplines and the Impact of Potential Biases. Front Psychol. 2019;10:813.

p. Gignac GE, Szodorai ET. Effect size guidelines for individual differences researchers. Personality and Individual Differences. 2016;102:74-8

---

## [Decision Letter · Decision Letter 1]

1 Sep 2022

PONE-D-22-06141R1Attention deficit in children with attention deficit hyperactivity disorder at primary school age measured with the attention network test (ANT): a protocol for a systematic review and meta-analysisPLOS ONE

Dear Dr. Gradys,

Thank you for submitting your manuscript to PLOS ONE. After careful consideration, we feel that it has merit but does not fully meet PLOS ONE’s publication criteria as it currently stands. Therefore, we invite you to submit a revised version of the manuscript that addresses the points raised during the review process.

The manuscript has been evaluated by two reviewers and their comments are attached below. They are satisfied with you revisions, but would like to clarify one minor point about which kind of abstracts were used.Could you please include this request before you resubmit your manuscript?

We look forward to receiving your revised manuscript.

Kind regards,

Thomas Tischer

Staff Editor

PLOS ONE

Journal Requirements:

Reviewers' comments:

Reviewer's Responses to Questions

**Comments to the Author**

1. Does the manuscript provide a valid rationale for the proposed study, with clearly identified and justified research questions?

Reviewer #1: Yes

Reviewer #2: Yes

2. Is the protocol technically sound and planned in a manner that will lead to a meaningful outcome and allow testing the stated hypotheses?

Reviewer #1: Yes

Reviewer #2: Yes

3. Is the methodology feasible and described in sufficient detail to allow the work to be replicable?

Reviewer #1: Yes

Reviewer #2: Yes

4. Have the authors described where all data underlying the findings will be made available when the study is complete?

Reviewer #1: Yes

Reviewer #2: Yes

5. Is the manuscript presented in an intelligible fashion and written in standard English?

Reviewer #1: Yes

Reviewer #2: Yes

6. Review Comments to the Author

You may also provide optional suggestions and comments to authors that they might find helpful in planning their study.

Reviewer #1: This is a very interesting study and a relevant topic for the scientific community. In addition, the authors have made all the requested changes.

Reviewer #2: The authors have carefully addressed all suggestions. They even have added information (e.g. data extraction) that properly complemented the protocol.

Just one minor issue regarding my question about meeting abstracts. Some databases retrieve abstracts from conferences or international meetings, which sometimes provide data. I consider it is important to state if only abstracts from published, peer review papers will be selected or also abstracts from conferences.

7. PLOS authors have the option to publish the peer review history of their article (what does this mean?). If published, this will include your full peer review and any attached files.

Reviewer #1: No

Reviewer #2: **Yes: **Yaira Chamorro

---

## [Author Response · Author response to Decision Letter 1]

19 Sep 2022

PONE-D-22-06141

RESPONSE TO REVIEWER COMMENTS

TITLE: Attention deficit in children with attention deficit hyperactivity disorder at primary school age measured with the attention network test (ANT): a protocol for a systematic review and meta-analysis

REVIEWER 1

Reviewer #1: This is a very interesting study and a relevant topic for the scientific community. In addition, the authors have made all the requested changes

RESPONSE: Thank you for the evaluation of our revision. Your kind words are deeply appreciated.

REVIEWER 2

Reviewer #2: The authors have carefully addressed all suggestions. They even have added information (e.g. data extraction) that properly complemented the protocol.

Just one minor issue regarding my question about meeting abstracts. Some databases retrieve abstracts from conferences or international meetings, which sometimes provide data. I consider it is important to state if only abstracts from published, peer review papers will be selected or also abstracts from conferences.

RESPONSE: We would like to thank you for your time to review our revised manuscript and for your kind words. Although we had considered it, due to the possibility of gathering more data, we decided not to include the conferences' abstracts. The conferences or international meetings' abstracts, even if containing information about the results, are not providing enough information to verify their study. Furthermore, that was repeatedly proven, that data from the conference abstracts cannot be trusted and the quality of their studies is hard to estimate . 

For clarification, in our manuscript, under the Search Strategy section we edited the following sentence: 

“The search will be not restricted to any language, sample size, or year of publication. We will exclude editorials, letters, case studies, case series, and conference abstracts.”

---

## [Editor Report · Decision Letter 2]

20 Sep 2022

Attention deficit in children with attention deficit hyperactivity disorder at primary school age measured with the attention network test (ANT): a protocol for a systematic review and meta-analysis

PONE-D-22-06141R2

Dear Dr. Gradys,

We’re pleased to inform you that your manuscript has been judged scientifically suitable for publication and will be formally accepted for publication once it meets all outstanding technical requirements.

Kind regards,

Thomas Tischer

Staff Editor

PLOS ONE
---

## [Editor Report · Acceptance letter]

5 Oct 2022

PONE-D-22-06141R2 

Attention deficit in children with attention deficit hyperactivity disorder at primary school age measured with the attention network test (ANT): a protocol for a systematic review and meta-analysis 

Dear Dr. Gradys:

I'm pleased to inform you that your manuscript has been deemed suitable for publication in PLOS ONE. Congratulations! Your manuscript is now with our production department. 

Kind regards, 

on behalf of

Dr. Thomas Tischer 

Staff Editor

PLOS ONE